# SimBench: Benchmarking the Ability of Large Language Models to Simulate Human Behaviors

## Abstract

Simulations of human behavior based on large language models (LLMs) have the potential to revolutionize the social and behavioral sciences, *if and only if* they faithfully reflect real human behaviors. Prior work across many disciplines has evaluated the simulation capabilities of specific LLMs in specific experimental settings, but often produced disparate results. To move towards a more robust understanding, we introduce SimBench, the first large-scale benchmark to evaluate how well LLMs can simulate group-level human behaviors across diverse settings and tasks. SimBench compiles 20 datasets in a unified format, measuring diverse types of behavior (e.g., decision-making vs. self-assessment) across hundreds of thousands of diverse participants from different parts of the world. Using SimBench, we can ask fundamental questions regarding when, how, and why LLM simulations succeed or fail. For example, we show that, while even the best LLMs today have limited simulation ability, there is a clear log-linear scaling relationship with model size, and a strong correlation between simulation and scientific reasoning abilities. We also show that base LLMs, on average, are better at simulating high-entropy response distributions, while the opposite holds for instruction-tuned LLMs. By making progress measurable, we hope that SimBench can accelerate the development of better LLM simulators in the future.

We combine **20 datasets** in a unified format.

| | |
|---|---|
| ChaosNLI | MoralMachineC |
| Choices13k | AfroBarometer |
| OpinionQA | OSPsychBig5 |
| NumberGame | DICES990 |
| WisdomOfCrowds | Jester |
| LatinoBarometro | ISSP ⋯ |

A train will kill 5 people on the track. You can flip a switch to divert the train to a side track where it will kill just 2 people.

**What do you do?**
**A**: Flip the switch
**B**: Do nothing

Each dataset contains **multiple-choice questions**.

We test the ability of LLMs to simulate **group-level responses**.

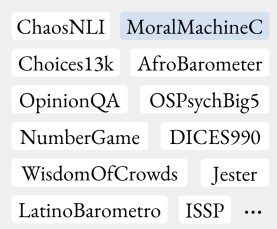
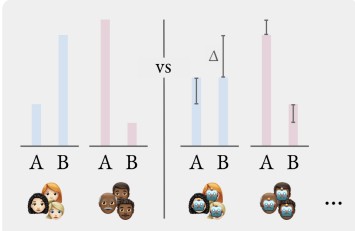

Figure 1: **SimBench** is the first-large scale benchmark to evaluate how well LLMs can simulate group-level human behavior across diverse simulation settings and tasks.

# 1 Introduction

Large-scale human experiments and surveys have long been essential tools for informing public policy, commercial decisions, and academic research. Running experiments and surveys, however, is costly and time-consuming. Large language models (LLMs) can potentially address this challenge by simulating human behaviors quickly and at low cost, to complement or even substitute human studies. This prospect, alongside encouraging early evidence on the efficacy of LLMs as simulators (Aher et al., 2023; Argyle et al., 2023; Horton, 2023), has motivated a large body of recent work across many disciplines investigating

the ability of LLMs to simulate human behaviors (Binz et al., 2024; Bisbee et al., 2024; Dominguez-Olmedo et al., 2024; Manning et al., 2024; Hu & Collier, 2025, inter alia).

Most prior work, however, has been highly specific, evaluating the simulation ability of a narrow set of LLMs for a specific set of tasks, producing varied and sometimes even conflicting results (§5). Overall, the evidence on LLM simulation ability resembles an incomplete patchwork, making it difficult to draw any broader conclusions about when, how, and why LLM simulations fail, or how LLMs can be trained to be better simulators.

To remedy these issues and enable a more robust science of LLM simulation, we introduce SimBench, the first large-scale benchmark for evaluating the ability of LLMs to simulate human behaviors across diverse settings and tasks. SimBench combines 20 datasets in a unified and easily adaptable format, including popular datasets used in prior work as well as new datasets used for the first time (Figure 1). Together, these datasets measure the ability of LLMs to simulate several distinct types of human behavior (e.g., decision-making vs. self-assessment) across a diversity of human respondents (e.g., from different parts of the world). With SimBench, we take a first step towards answering six fundamental research questions about the simulation ability of LLMs:

> **RQ1**: How well can current LLMs simulate human behaviors across diverse settings and tasks?

We test 24 state-of-the-art LLMs (§3), and show that even the best LLMs today struggle to faithfully simulate group-level human behaviors (§4.1). Predictions from the best-performing LLM, on average, are closer to a uniform response baseline than the true human response distribution.

> **RQ2**: How do LLM characteristics such as model size affect LLM simulation ability?

We show that simulation ability grows log-linearly with model size (§4.2). We also find indicative evidence that increasing test-time compute does not meaningfully improve LLM simulations.

> **RQ3**: How does task selection affect LLM simulation fidelity?

We find that simulation fidelity varies substantially across tasks, with even the best LLM simulators consistently performing worse than a uniform response baseline on several datasets (4.3).

> **RQ4**: How does the degree of human response plurality affect LLM simulation fidelity?

We find that instruction-tuned LLMs tend to perform better on questions where humans give similar answers whereas base LLMs tend to perform better on questions where humans differ (§4.4).

> **RQ5**: Are LLMs better at simulating responses from some groups than others?

We show that, on SimBench, LLMs struggle more with simulating specific demographic groups, especially those based on religion and ideology, compared to general populations (§4.5).

> **RQ6**: To what extent does LLM simulation ability correlate with different model capabilities?

We find positive correlations with several popular capability benchmarks, including a particularly strong correlation with performance on scientific reasoning tasks (§4.6).

Progress in AI is only possible through rigorous evaluation, and large-scale benchmarks such as MMLU (Hendrycks et al., 2021) have significantly contributed to improvements in LLM capabilities. We hope that SimBench can play a similar role in accelerating the development of LLMs for simulating human behaviors. All of SimBench is permissively licensed and available on GitHub and Hugging Face.

## 2 Creating SimBench

### 2.1 Selecting Datasets for SimBench

To create SimBench, we conducted an open-ended search for suitable datasets in the social and behavioral sciences, guided by two main selection criteria: i) **large participant counts**, so that each dataset captures meaningful response distributions rather than the idiosyncratic behavior of few individuals; and ii) **permissive licensing** to freely redistribute each dataset as part of SimBench.

We generally opted for **datasets that have not been used to evaluate LLMs** in prior work, to increase the novelty and effectiveness of SimBench. However, to increase coverage and backward comparability, we also included datasets used in prior work (e.g., OpinionQA, ChaosNLI).

We also prioritized **datasets that provide participants' sociodemographic information** to evaluate the ability of LLMs to simulate responses from specific participant groups (see §2.3). Most survey datasets, for example, include this information. However, we also included three datasets that do not provide sociodemographic information (Jester, ChaosNLI, Choices13k) because they substantially increase the overall task diversity in SimBench.

Overall, SimBench includes 20 datasets, which we list in Appendix G, providing details on participants and example questions. Crucially, SimBench is fully modular by design, so that future work can easily add more datasets using the processing pipeline described in §2.2 below. In its release version, SimBench already meets two key criteria for comprehensive evaluation of LLM simulation ability:

1) **Task Diversity**: The 20 datasets in SimBench cover a wide range of different tasks regarding the human behavior they measure. SimBench includes **decision-making** questions (e.g., in Choices13k, MoralMachine), where participants are presented with a set of actions that concern themselves, and they have to select the action they would hypothetically take. SimBench also includes **self-assessment** questions (e.g., in OpinionQA, OSPsychBig5), where participants are presented with a set of descriptions or attributes, and they have to select the one that best describes themselves. Further, SimBench includes **judgment** questions (e.g., in ChaosNLI and Jester) where participants are presented with some external object and a choice of labels, and they have to select the label they think fits best. Lastly, SimBench includes **problem-solving** questions (e.g., in WisdomOfCrowds and OSPsychMGKT), where participants are presented with a set of answers to a factual question, and they have to select the answer they think is correct. Consequently, LLMs have to accurately simulate several distinct types of human behavior in order to perform well on SimBench.

2) **Participant Diversity**: The 20 datasets in SimBench capture a rich demographic landscape spanning at least 130 different countries across six continents. This global representation is a key strength of the benchmark. While five datasets include US-based crowdworkers, the international scope of SimBench is substantial: 3 datasets (e.g., LatinoBarometro, Afro-Barometer) exclusively feature participants from regions outside the US, 4 datasets (e.g., GlobalOpinion, TISP) draw from multi-country samples across different continents, and 2 datasets collect responses from a global pool of internet users. Importantly, 8 out of the 20 datasets employ representative sampling techniques, enhancing the ecological validity of these constituent components. To perform well on SimBench, LLMs must therefore demonstrate the ability to accurately simulate the behavior of human participants across diverse cultural, linguistic, and socioeconomic backgrounds.[1]

### 2.2 Unifying SimBench Dataset Formats

**Question Selection & Format:** SimBench is a multiple-choice benchmark. From all 20 datasets, we therefore select only multiple-choice questions, and transform continuous scale

---

[1]Note that, while some constituent datasets recruit representative samples, SimBench as a whole is not fully representative of any specific group of participants.

questions into multiple-choice by splitting the scale into uniform bins. Where applicable, we collapse answer options to limit the maximum number of answering options to at most 26. In practice, questions rarely have more than 11 options. We exclude any questions with free-text answers and questions that are contingent on prior questions or with multi-turn interactions. For datasets with questions that are not originally in English, we use the English-language equivalents provided by the dataset creators. We do this to enable consistent evaluation, but we note that simulation ability may plausibly be correlated with prompt language, and encourage future work in this direction.

**Grouping Variables:** For each dataset, we record a brief description of the overall sampling population, the *default grouping*, in the form of a short prompt. For example, all participants in the WisdomOfCrowds dataset were US-based Amazon Mechanical Turk workers, so the default grouping prompt for this dataset is "You are an Amazon Mechanical Turk worker based in the United States.". Additionally, we select *grouping variables* for each dataset, corresponding to known participant sociodemographics, like age, gender, or race. The exact grouping variables and their values depend on what is available for each dataset. For a list of all grouping variables for each dataset, see Appendix G.

**Response Distributions:** We record the answers to each question in SimBench as group-level response distributions over the question's multiple-choice options. These distributions serve as the reference that we compare LLM predictions to. We create group-level response distributions by aggregating over the answers from all participants that belong to a given group. We set minimum grouping size thresholds for each dataset, filtering out groups with insufficient participants to form meaningful response distributions. Through this aggregation process, SimBench encompasses **10,930,271** unique question, grouping variable value pairs, each representing a distinct simulation target (see Table 3 for detailed counts). This approach enables robust evaluation of how accurately LLMs can simulate response patterns across diverse demographic groups and question types.

## 2.3 SimBench Splits

While the complete SimBench contains over 10 million potential test cases, for practical evaluation purposes we focus on two carefully curated splits that still provide comprehensive coverage of the simulation capabilities we aim to assess:

1) The **SimBenchPop** split covers all questions in all 20 datasets after processing as in §2.2. We combine each question with the dataset-specific default grouping prompt to create one unique test case, resulting in 7,167 test cases. We obtain the response distribution for each test case by aggregating all individual responses to that test case over all participants in that dataset. Conceptually, **SimBenchPop measures the ability of LLMs to simulate responses of broad and diverse human populations**.

2) The **SimBenchGrouped** split contains only the five large-scale survey datasets in SimBench (AfroBarometer, ESS, ISSP, LatinoBarometro, and OpinionQA) because for these datasets we have enough participants to obtain meaningful group sizes even when selecting on a specific group attribute (e.g., age = 30-49). For each dataset, we select questions that exhibit significant variation across demographic groups, ensuring that the benchmark captures meaningful demographic differences in responses. This results in 6,343 test cases overall. For more details on the sampling process, see Appendix D. Conceptually, **SimBenchGrouped measures the ability of LLMs to simulate responses from narrower participant groups based on specified group characteristics**.[2]

---

[2]Ideally, we would also like to measure LLM simulation ability for intersectional groups that combine multiple characteristics (e.g., female + age 30-49). However, selecting on multiple characteristics substantially decreases group size, thus increasing sampling noise in the response distributions. Reliable evaluation of intersectional group simulation ability would require datasets with more participants than we have access to.

## 3 Experimental Setup

**Tested Models:** To demonstrate the usefulness of SimBench and answer our six research questions (§1), we evaluate 24 state-of-the-art LLMs across 7 model families on SimBench. This includes both commercial and open-weight, base and instruction-tuned models, with model sizes ranging from 0.5B to 405B parameters. Table 1 shows the full list of models.

**Model Elicitation:** For each model, we collect predictions for the two main splits of Sim-Bench (§2.3). To obtain model response distributions, we use one of two methods, depending on model type: 1) For base models, we directly extract **token probabilities** for each response option based on first-token logits. This is a natural way of eliciting a distribution out of an LLM, especially a base LLM. 2) For instruction-tuned models, we follow recent literature on LLM calibration and distribution prediction (Tian et al., 2023; Meister et al., 2025) and use **verbalized distributions**, e.g., "Option A: 30%, Option B: 70%", elicited through prompting. For implementation details and prompt formats, see Appendix I.

**Evaluation Metric**: To measure LLM simulation ability, we derive the SimBench score $S$ from Total Variation Distance TVD, defined as:

$$S(P,Q) = 100 \left(1 - \frac{TVD(P,Q)}{TVD(P,U)}\right) = 100 \left(1 - \frac{\sum_i |P_i - Q_i|}{\sum_i |P_i - U_i|}\right) \tag{1}$$

where $P$ is the human ground truth distribution, $Q$ is the distribution predicted by the LLM that is being tested, and $U$ is a uniform distribution over all response options for a given question. Conceptually, $S$ therefore measures how much more accurate the predictions from an LLM are than predictions from a uniform baseline model, which assigns equal probability to all response options for a given question. In other words, $S$ quantifies the advantage of an LLM simulation over the simplest possible guess.

An $S$ score of 100 indicates perfect alignment between the LLM and the human ground truth distribution, while a score ≤0 indicates performance at or below the performance of a uniform baseline. We chose TVD as the basis for $S$ due to its symmetry, boundedness, and robustness to zero probabilities. For a comparison to alternative metrics, see Appendix E.

Table 1: **Overall simulation ability** as measured by SimBench score $S$ averaged across the two main splits of SimBench. Reasoning models are highlighted in *italics*. Models are sorted by score. Models below the dotted line perform worse than a uniform baseline.

| Model | Type | Release | S (↑) |
|---|---|---|---|
| Claude-3.7-Sonnet | Instr. | Closed | 40.80 |
| *Claude-3.7-Sonnet-4000* | Instr. | Closed | 39.46 |
| GPT-4.1 | Instr. | Closed | 34.56 |
| *DeepSeek-R1* | Instr. | Open | 34.52 |
| DeepSeek-V3-0324 | Instr. | Open | 32.90 |
| *o4-mini-high* | Instr. | Closed | 28.99 |
| Llama-3.1-405B-Instruct | Instr. | Open | 28.41 |
| *o4-mini-low* | Instr. | Closed | 27.77 |
| Gemma-3-12B-IT | Instr. | Open | 18.63 |
| Gemma-3-27B-IT | Instr. | Open | 18.34 |
| Llama-3.1-70B-Instruct | Instr. | Open | 16.57 |
| Qwen2.5-72B | Base | Open | 13.35 |
| Qwen2.5-32B | Base | Open | 12.28 |
| Qwen2.5-14B | Base | Open | 11.93 |
| Qwen2.5-3B | Base | Open | 8.84 |
| Qwen2.5-7B | Base | Open | 8.76 |
| Gemma-3-12B-PT | Base | Open | 7.67 |
| Gemma-3-27B-PT | Base | Open | 5.54 |
| Qwen2.5-1.5B | Base | Open | 5.34 |
| Llama-3.1-8B-Instruct | Instr. | Open | -0.14 |
| Gemma-3-4B-PT | Base | Open | -0.73 |
| Gemma-3-4B-IT | Instr. | Open | -1.91 |
| Qwen2.5-0.5B | Base | Open | -2.99 |
| Gemma-3-1B-PT | Base | Open | -16.13 |

## 4 Results

### 4.1 RQ1:
### General Simulation Ability of LLMs

To evaluate the general simulation ability of LLMs, we measure their overall Sim-Bench score $S$ averaged across the two main splits of SimBench (Table 1). We find that **even leading LLMs struggle to simulate group-level human behaviors with high accuracy**, as measured across the 20 datasets in SimBench. Claude-3.7-Sonnet is the best-performing model overall, but only achieves a score of 40.80 out of a maximum of 100 on SimBench. This score indicates that the

response distributions predicted by Claude-3.7-Sonnet are, on average, closer to a uniform response distribution than to the true human response distribution. The distance from the true distribution is 19.7 percentage points, on average, as shown by the TVD listed in Table 5. The best-performing open-weight LLM is DeepSeek-R1, achieving a score of 34.52. The majority of the 24 models we test perform substantially worse still, scoring less than 20. Notably, five models we test score below 0, indicating that their predicted response distributions are, on average, even further away from the true human response distribution than a uniform response distribution. Overall, these results suggest that disparate results from prior work may combine into a somewhat disappointing picture, painting LLMs as far from reliable simulators when considering a diversity of tasks.

## 4.2 RQ2: Impact of LLM Characteristics on Simulation Ability

While even the best models struggle to perform well on SimBench, Table 1 also shows clear differences across models. Therefore, we investigate how performance varies depending on model characteristics, specifically 1) model size, and 2) test-time compute.

**1) Model Size** To evaluate the impact of model size on simulation ability, we plot SimBench Score $S$ against model parameter count for the four LLM families that we can test across multiple model sizes (Figure 5). Our results suggest that **there is a clear log-linear scaling law for LLM simulation ability**. Across all examined model families, an increase in parameter count generally corresponds to an increase in SimBench score $S$, indicating better alignment between predicted and human response distributions. Llama-3.1-Instruct in particular demonstrates nearly perfect log-linear scaling, with the largest Llama-3.1-405B-Instruct achieving a score of 28.41. Conversely, all models with low parameter counts ($\leq$10B) perform very poorly on SimBench, scoring at most 8.76 (Qwen2.5-7B). Overall, the clear positive scaling trends across model families suggest that, while simulation remains a challenging task for even the best models today, further model scaling may well lead to highly accurate LLM simulators in the future.

**2) Test-Time Compute** To analyze the effects of increasing test-time compute on LLM simulation ability, we compare o4-mini-low vs. o4-mini-high, as well as Claude-3.7-Sonnet in its standard configuration vs. with a 4000-token thinking budget (Table 1). We are limited to these two comparisons due to budget constraints. Our results suggest that **there is no clear benefit to increasing test-time compute for LLM simulation ability**. However, this finding should only be interpreted as early, indicative evidence, and we hope that SimBench can enable further work in this direction.

## 4.3 RQ3: Impact of Task Selection on Simulation Fidelity

The 20 datasets in SimBench correspond to very different tasks, in terms of the aspects of human behavior that they measure (see §2.1). Therefore, we break down simulation fidelity by dataset, showing results for the five LLMs we previously identified as the best simulators in Figure 2. We find that **simulation fidelity varies substantially across tasks**, with even the best LLM simulators performing worse than a uniform response baseline on several datasets, as indicated by negative SimBench scores (e.g., on Jester, OSPsychMach, and MoralMachine). Generally, the different LLMs exhibit similar performance patterns, with one notable exception being GPT-4.1's exceptionally high score of 61.9 on OSPsychRWAS.

## 4.4 RQ4: Impact of Response Plurality on Simulation Fidelity

Human participants give very similar responses to some questions while giving very different responses to others. Faithful simulation requires models to perform well in either scenario. We operationalise the level of response plurality by measuring the normalised entropy of the human response distribution at the question level.

We then plot this entropy for all questions in SimBenchPop against to-tal variation distance (TVD, see §3), which measures the difference in predicted and reference distribution at a question level (Figure 3).

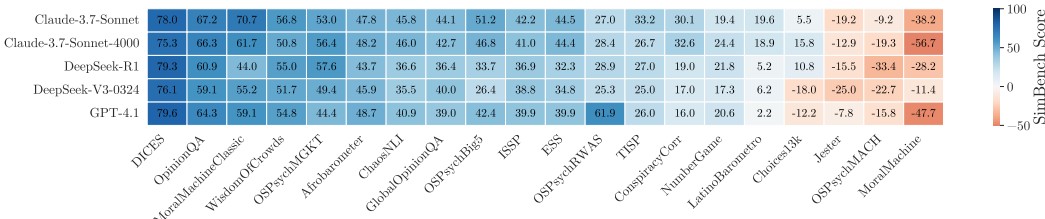

Figure 2: **Simulation fidelity by dataset** as measured by SimBench score $S$ for each of the 20 datasets in SimBenchPop. We show results for the top five models based on results in Table 1.

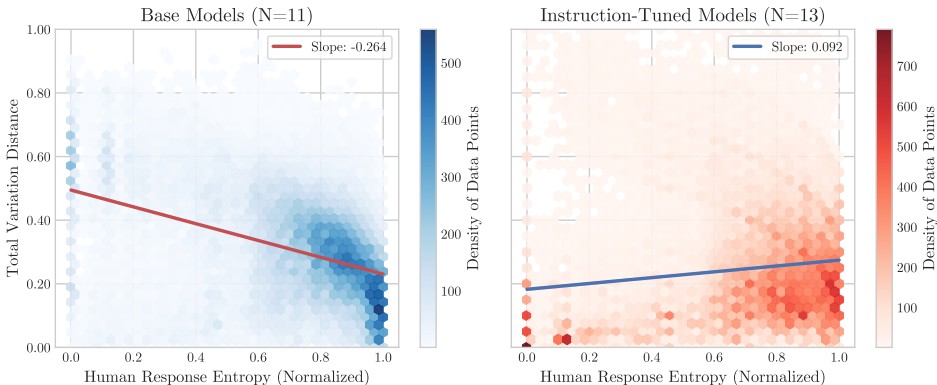

Figure 3: **Response plurality vs. simulation fidelity** for base and instruction-tuned models on all questions in SimBenchPop. We measure response plurality by normalised entropy of the human response distribution and simulation fidelity by total variation distance at the question level.

Prior work has found that instruction-tuning encourages models to produce more confident, less ambiguous outputs, resulting in low-entropy token distributions (Brown et al., 2020; Tian et al., 2023; Meister et al., 2025; Cruz et al., 2024). Therefore, we differentiate between base and instruction-tuned models for this analysis. We find that **base models generally perform better on questions where human participants tend to give different answers, whereas the inverse holds for instruction-tuned models**. This finding is supported by our regression analysis in Appendix 6, which confirms the statistical significance of this effect. Therefore, while instruction-tuned models tend to outperform base models in terms of overall score on SimBench (Table 1), our results here suggest that instruction-tuning also worsens simulation ability for at least a subset of high-plurality questions.

### 4.5 RQ5: Simulation Ability Across Participant Groups

Many applications require simulating responses from specific demographic groups rather than general populations. Using SimBenchGrouped, we evaluate how LLM simulation ability changes when conditioned on specific demographic attributes.

Table 2: **Ungrouped vs. grouped** simulation performance $\Delta S$.

| Models | |
|---|---|
| Claude-3.7-Sonnet | -3.13 |
| Claude-3.7-Sonnet-4000 | -4.61 |
| DeepSeek-R1 | -3.79 |
| DeepSeek-V3-0324 | -1.27 |
| GPT-4.1 | -3.94 |

| Demographics | |
|---|---|
| Religiosity/Practice | -9.91 |
| Political Affil./Ideology | -4.97 |
| Religion (Affiliation) | -4.83 |
| Income/Social Standing | -4.51 |
| Domicile/Urbanicity | -3.17 |
| Employment Status | -3.03 |
| Education | -2.55 |
| Marital Status | -1.80 |
| Age | -1.50 |
| Gender | -1.24 |

We measure this change as $\Delta S = S_{grouped} - S_{ungrouped}$, where $S_{ungrouped}$ is the SimBench score for simulating the general population and $S_{grouped}$ is the score when simulating a specific demographic group on the same question. A negative $\Delta S$ indicates that the model's simulation ability relative to the uniform baseline decreases when asked to simulate specific demographic groups.

Importantly, for SimBenchGrouped, we specifically selected questions where human response distributions showed the highest variance across demographic groups (see §2.3). The observed degradation in simulation performance therefore likely represents an upper bound on the challenges LLMs face when simulating specific demographic groups. Our results in Table 2 show that **LLMs struggle more with simulating specific demographic groups compared to general populations**. All evaluated models show negative mean $\Delta S$ values, with degradation ranging from -1.27 for DeepSeek-V3-0324 to -4.61 for Claude-3.7-Sonnet-4000.

The performance degradation varies substantially by demographic category. Models struggle most when simulating groups defined by religious attributes, with conditioning on 'Religiosity/Practice' causing the largest decrease in simulation accuracy ($\Delta S = -9.91$), followed by 'Political Affiliation/Ideology' ($\Delta S = -4.97$) and 'Religion (Affiliation)' ($\Delta S = -4.83$). In contrast, models maintain relatively better performance when simulating groups defined by 'Gender' ($\Delta S = -1.24$) and 'Age' ($\Delta S = -1.50$).

While these findings may not fully generalize to cases where demographic differences are less pronounced, they highlight potential limitations in how current LLMs capture the nuanced response patterns of specific demographic groups. We argue that such challenging benchmarks are crucial for identifying areas where improvements are most needed, particularly for applications that aim to model the behaviors of specific subpopulations.

## 4.6   RQ6: Simulation Ability vs. General Capabilities

Finally, we analyze the relationship between LLM simulation ability and more general model capabilities by correlating performance on SimBench with popular LLM capability benchmarks (Figure 4). Specifically, we compare SimBench scores to performance on GPQA Diamond (Rein et al., 2024) and OTIS AIME (EpochAI, 2024), based on scores reported in the Epoch AI Benchmarking Hub (Epoch AI, 2024), which we are able to retrieve for 8 of the LLMs we test. We also compare to Chatbot Arena ELO scores (Chiang et al., 2024), retrieved for the same 8 models on May 14th, 2025.

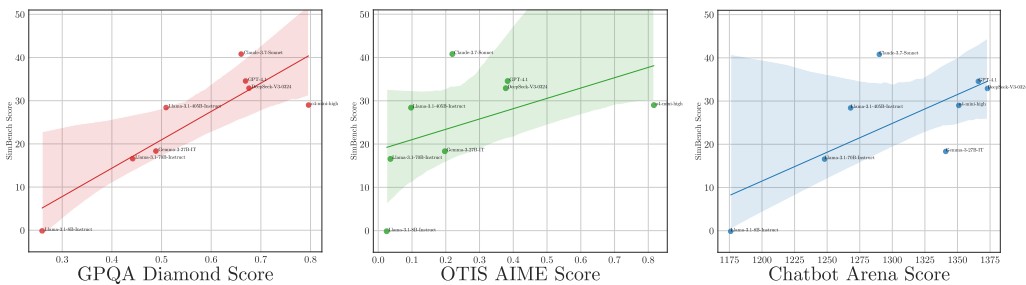

Figure 4: **General model capabilities vs. simulation ability**, as measured by popular benchmark scores compared to SimBench score $S$ averaged across the two main splits in SimBench.

We find that **simulation ability is positively correlated with general model capabilities**. This matches our earlier finding on the benefits of model scaling (§4.2). However, the strength of the correlation varies across capability benchmarks. Most notably, the very strong correlation with GPQA suggests that there may be substantial symbiotic effects between scientific reasoning and simulation for social and behavioral science tasks of the kind included in SimBench. By comparison, the weaker correlation with Chatbot Arena scores suggests optimising LLMs for general helpfulness and user satisfaction does not necessarily make them better simulators.

## 5 Related Work

**Human Behavior Simulation with LLMs**  LLMs as human behavior simulators have attracted significant interdisciplinary attention. Researchers have evaluated their efficacy across political science (Argyle et al., 2023; Bisbee et al., 2024; Dominguez-Olmedo et al., 2024), psychology (Aher et al., 2023; Binz et al., 2024; Manning et al., 2024; Hewitt et al., 2024), economics (Horton, 2023; Aher et al., 2023), and computer science applications (Hu & Collier, 2024; Dong et al., 2024; Hu & Collier, 2025; Park et al., 2023). Evidence regarding LLMs' simulation fidelity remains mixed, with some studies reporting promising results (Argyle et al., 2023) while others identify critical limitations, including homogenized group representations (Cheng et al., 2023; Wang et al., 2025) and deterministic rather than distributional predictions (Park et al., 2024b).

Existing work has predominantly focused on individual-level simulation with minimal demographic conditioning, typically evaluating only one or two models in narrowly defined contexts. SimBench addresses these limitations by providing a comprehensive benchmark for group-level simulation across diverse domains with systematic demographic conditioning and standardized metrics. The benchmark's distributional evaluation framework (using Total Variation distance) captures how accurately models represent the full spectrum of human response variation—an approach advocated by researchers in both simulation (Anthis et al., 2025) and general LLM evaluation (Ying et al., 2025). For broader context on this emerging field, we refer readers to recent comprehensive surveys (Kozlowski & Evans, 2024; Olteanu et al., 2025; Anthis et al., 2025).

**Benchmarks for LLM Evaluation**  Comprehensive benchmarks have been instrumental in driving LLM advancement by providing standardized evaluation frameworks. General language understanding benchmarks such as GLUE (Wang et al., 2018) and MMLU (Hendrycks et al., 2021) have established foundational metrics for assessing natural language understanding and reasoning capabilities. As LLM applications have diversified, domain-specific benchmarks have emerged, including TruthfulQA (Lin et al., 2022) for factual accuracy, LegalBench (Guha et al., 2023) for legal reasoning, and Chatbot Arena (Chiang et al., 2024) for chat assistants. These specialized benchmarks have enabled more precise evaluation of LLMs' fitness for particular use cases and have guided domain-specific optimization.

Appendix H continues our discussion of related work.

## 6 Conclusion

LLM simulations of human behavior have the potential to create immense benefits for society by helping shape effective policy, guiding industrial decisions, and informing academic research. To fulfill this potential, however, LLM simulations must be sufficiently faithful in representing real human behaviors across diverse settings and tasks. Prior work evaluating LLM simulation fidelity has taken a predominantly narrow approach, producing an incomplete patchwork of evidence.

To change this, we introduced SimBench, the first large-scale benchmark for evaluating group-level LLM simulation ability. We described the dataset selection and processing steps that resulted in 20 datasets with a unified format, measuring diverse types of human behavior (e.g., decision-making vs. self-assessment) across hundreds of thousands of diverse participants from different parts of the world. Using SimBench, we took a first step toward answering fundamental questions regarding when, how, and why LLM simulations succeed or fail. For example, we demonstrated that while even the best LLMs today have limited simulation ability, there is a clear log-linear scaling relationship with model size and a strong correlation between simulation and scientific reasoning abilities.

Significant progress remains to be made in developing LLMs as better simulators of human behavior. We hope that SimBench can provide an open foundation for future efforts in this direction, ultimately benefiting society as a whole.

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

## A   Limitations

**Scope of Representativeness**   Although SimBench spans 20 diverse datasets, the combined sample does (and can) not fully represent any single population in its full complexity. Many geographic regions are still underrepresented or entirely absent, potentially limiting generalizability to populations with different cultural backgrounds and preferences. Even within countries, demographic representativeness may vary, as only a subset of our 20 datasets are based on nationally representative sampling techniques. Each dataset carries its own statistical uncertainty. Opt-in samples and crowdsourced data (e.g., from Amazon Mechanical Turk) may have larger margins of error than nationally representative surveys, potentially affecting the benchmark's precision for certain questions. We view these limitations as opportunities for collaborative extension of SimBench to improve global coverage and representativeness over time.

**Temporal Dimensions**   The current version of SimBench utilizes static datasets that capture human behavior at specific points in time. This approach allows for systematic evaluation across domains but cannot yet assess how well LLMs simulate evolving preferences, opinion shifts, or behavioral adaptation—all fundamental aspects of human behavior. Future iterations of SimBench could incorporate longitudinal data to address these dynamic aspects of human behavior and expand the benchmark's evaluative capacity.

**Task Format Considerations**   SimBench currently focuses on multiple-choice, single-answer, single-turn questions and interactions. This standardized format enables systematic comparison across diverse domains but necessarily excludes more complex behavioral simulations including multi-step decision processes and interactive social dynamics. We see this as a pragmatic starting point that establishes foundational evaluation capabilities while inviting future extensions to capture more nuanced aspects of human behavior.

**Training Data Overlap**   Without complete transparency into model training corpora, we cannot definitively rule out the possibility that some test items appeared during training. However, several factors mitigate concerns about data contamination affecting our results. First, SimBench evaluates simulation at the group distribution level rather than individual response prediction, making memorization of specific survey responses less impactful. Second, many of our datasets primarily exist as aggregated statistics in published research rather than as widely available raw data. Finally, the consistent scaling patterns we observe across diverse datasets suggest genuine simulation capabilities rather than artifacts of training data overlap. Nevertheless, we acknowledge that data contamination remains a fundamental challenge in LLM evaluation, and future work should develop more robust methods to detect and quantify its impact. We include this consideration for completeness while believing it unlikely to significantly impact our current findings.

## B   Ethical Considerations

SimBench's primary purpose is to benchmark LLMs' ability to simulate human behavior. While advancements in LLM simulation capabilities can support helpful applications such as pre-testing policies, these do not come without risks of misrepresentation and dual use.

First and foremost, due to the observed limited simulation ability of state-of-the-art LLMs, we caution against relying on LLM-powered simulations of human behavior for tasks where downstream harm is possible. Even as models improve, substituting algorithmic approximations for authentic human participation carries the risk of disadvantaging under-represented/marginalized communities by removing their opportunities to directly shape decisions that affect them. Furthermore, while benchmarks like SimBench help measure simulation capabilities, we must be careful not to mistake increasing benchmark performance for genuine understanding of complex human behavior.

While SimBench includes diverse demographic groups, it can not adequately support simulations of intersectional identities due to sample size limitations. By conditioning on one

demographic variable at a time, we cannot systematically assess how well models handle the rich overlap of identities (e.g., "older Latinx women," "young Black men"). Small intersectional group sizes make it difficult to combine multiple characteristics simultaneously due to increasing sampling noise in response distributions. Yet intersectional simulation is precisely where societal biases and model limitations often emerge, making this an important direction for future work. Additionally, the conditional prompting approach we use conceptualizes simplistic human populations and may thus fail to appropriately account for nuances of individual behavior.

Nevertheless, we believe SimBench is an important step toward making LLM simulation progress measurable and raising awareness of state-of-the-art model blind spots. Together, we hope this will ultimately create accountability for models deployed in socially sensitive contexts.

## C   Scaling Results

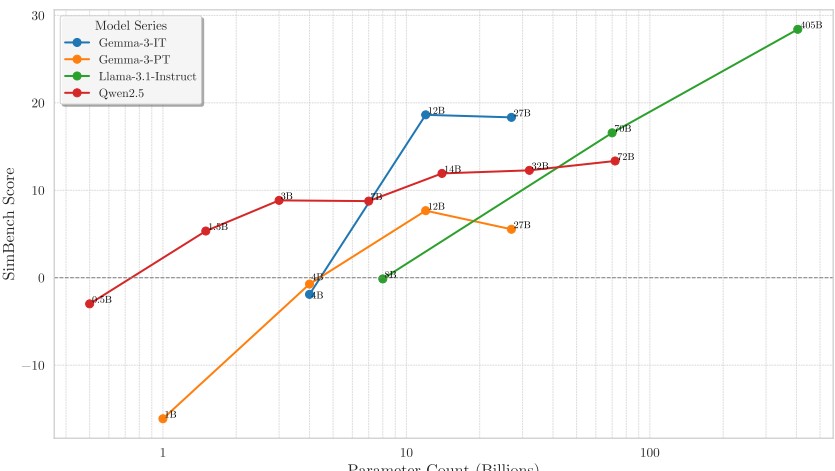

Figure 5: **Model parameter count vs. simulation ability**. We measure model size by parameter count and simulation ability by SimBench score *S* averaged across the two main splits of SimBench.

## D   SimBenchPop and SimBenchGrouped Sampling Details

We curated data at two levels of grouping granularity, corresponding to our two main benchmark splits: **SimBenchPop** and **SimBenchGrouped**.

**SimBenchPop** measures LLMs' ability to simulate responses of broad, diverse human populations. We include all questions from all 20 datasets in SimBench, combining each question with its dataset-specific default grouping prompt (e.g., "You are an Amazon Mechanical Turk worker based in the United States"). We sample up to 500 questions per dataset to ensure representativeness while keeping the benchmark manageable. For each test case, we aggregate individual responses across all participants in the dataset to create population-level response distributions. This approach creates a benchmark that represents population-level responses across diverse domains while maintaining a reasonable size of 7,167 test cases.

For **SimBenchGrouped**, we focus only on five large-scale survey datasets with rich demographic information and sufficient sample sizes: OpinionQA, ESS, Afrobarometer, ISSP, and LatinoBarometro. Our sampling approach prioritizes questions showing meaningful demographic variation. For each dataset, we identify available grouping variables (e.g., age, gender, country) with sufficient group sizes to form meaningful response distributions. We

calculate the variance of responses across demographic groups for each question and rank questions by their variance scores, prioritizing those showing the strongest demographic differences. We select questions that exhibit significant variation across demographic groups to ensure the benchmark captures meaningful differences in responses. For each selected question, we create multiple test cases by pairing it with different values of the grouping variables (e.g., age = "18-29", age = "30-49"). This process results in 6,343 test cases that specifically measure LLMs' ability to simulate responses from narrower participant groups based on specified demographic characteristics. Table 3 provides a summary of the sampling process across all datasets, showing the minimum group size thresholds and the number of test cases in each benchmark split.

Table 3: Dataset Sampling Summary; NaN refers to dataset that is only available in aggregated form and no grouping size is known.

| Dataset | Min. Group | SimBench | SimBenchPop | SimBenchGrouped |
|---|---|---|---|---|
| WisdomOfCrowds | 100 | 1,604 | 114 | – |
| Jester | 100 | 136 | 136 | – |
| Choices13k | NaN | 14,568 | 500 | – |
| OpinionQA | 300 | 1,074,392 | 500 | 984 |
| MoralMachineClassic | 100 | 3,441 | 15 | – |
| MoralMachine | 100 | 20,771 | 500 | – |
| ChaosNLI | 100 | 4,645 | 500 | – |
| ESS | 300 | 2,783,780 | 500 | 1,643 |
| Afrobarometer | 300 | 517,453 | 500 | 1,531 |
| OSPsychBig5 | 300 | 1,950 | 250 | – |
| OSPsychMACH | 300 | 3,682,700 | 100 | – |
| OSPsychMGKT | 300 | 20,610 | 500 | – |
| OSPsychRWAS | 300 | 975,585 | 22 | – |
| ISSP | 300 | 594,336 | 500 | 940 |
| LatinoBarometro | 300 | 80,684 | 500 | 1,245 |
| GlobalOpinionQA | NaN | 46,329 | 500 | – |
| DICES | 10 | 918,064 | 500 | – |
| NumberGame | 10 | 15,984 | 500 | – |
| ConspiracyCorr | 300 | 968 | 45 | – |
| TISP | 300 | 172,271 | 485 | – |
| **Total** | | **10,930,271** | **7,167** | **6,343** |

## E  Metric Robustness Check

TVD ranges from 0 (perfect match) to 1 (complete disagreement), with lower values indicating better simulation fidelity. TVD provides an interpretable measure of how closely model predictions align with actual human response distributions. TVD is particularly well-suited for simulation evaluation compared to alternatives like KL divergence or Jensen-Shannon divergence (JSD). Unlike KL divergence, TVD remains well-defined even when the model assigns zero probability to responses that humans give, avoiding the infinite penalties that KL would impose in such cases. Additionally, TVD is symmetric and bounded, making it more interpretable across different datasets and response distributions than KL divergence. While JSD offers similar advantages in terms of symmetry and boundedness, TVD provides a more direct and intuitive interpretation of the maximum possible error in probability estimates. This property is especially valuable when evaluating how accurately models simulate the distribution of human responses rather than just matching the most likely response. For further discussion on TVD as an evaluation metric, see also Meister et al. (2025). We show the results of Table 1 in terms of raw TVD values in Table 5.

To ensure our findings are robust across different metrics, we complement TVD with two alternative metrics: Jensen-Shannon Divergence (JSD) and Spearman's Rank Correlation

(RC). Table 4 presents these metrics for a subset of evaluated models. The strong Pearson correlation between TVD and JSD ($r = 0.92$) indicates these metrics provide consistent model rankings. The moderate negative correlation ($r = -0.57$) between TVD and RC is expected, as lower distances correspond to higher correlations. This multi-metric evaluation confirms that our model comparisons remain consistent across different statistical measures.

Table 4: Comparison of models on three metrics: Total Variation Distance (TVD), Jensen-Shannon Divergence (JSD), and Spearman Rank Correlation (RC). Lower values are better for TVD and JSD; higher is better for RC.

| Model | Total Variation | JS Divergence | Rank Correlation |
|---|---|---|---|
| Claude-3.7-Sonnet | 0.191 | 0.057 | 0.673 |
| Claude-3.7-Sonnet-4000 | 0.195 | 0.060 | 0.648 |
| DeepSeek-R1 | 0.211 | 0.069 | 0.623 |
| DeepSeek-V3-0324 | 0.216 | 0.069 | 0.620 |
| GPT-4.1 | 0.209 | 0.070 | 0.646 |
| Llama-3.1-405B-Instruct | 0.231 | 0.085 | 0.593 |
| o4-mini-high | 0.225 | 0.079 | 0.621 |
| o4-mini-low | 0.230 | 0.082 | 0.609 |

# F   Regression Analysis of Human Response Entropy and Model Performance

To formally test the relationship between human response entropy and simulation performance across different model types, we fit an Ordinary Least Squares (OLS) regression model predicting Total Variation (TV) distance at the individual question-model level. The model specification was as follows:

$$\text{Total\_Variation} \sim C(\text{dataset\_name}) + C(\text{model}) + C(\text{instruct\_flag}) : \text{Human\_Normalized\_Entropy} \tag{2}$$

Here, *Total_Variation* is the dependent variable. $C(\text{dataset\_name})$ and $C(\text{model})$ represent fixed effects for each dataset and model, respectively, controlling for baseline differences in difficulty and capability. The crucial term is the interaction $C(\text{instruct\_flag}) :$ Human_Normalized_Entropy, where *instruct_flag* is a binary indicator for instruction-tuned models (0 for base, 1 for instruction-tuned).

The key results from Table 6 are the coefficients for the interaction terms:

- For base models: The coefficient on the interaction between base models and Human Normalized Entropy is $-0.2555$ ($p < 0.001$), indicating that for every one-unit increase in normalized entropy, the TVD decreases by approximately 0.26 units. This means that base models perform *better* (lower TVD) when simulating human populations with more diverse opinions.

- For instruction-tuned models: The coefficient on the interaction between instruction-tuned models and Human Normalized Entropy is $+0.1072$ ($p < 0.001$), indicating that for every one-unit increase in normalized entropy, the TVD increases by approximately 0.11 units. This means that instruction-tuned models perform *worse* (higher TVD) when simulating human populations with more diverse opinions.

These coefficients are both highly statistically significant ($p < 0.001$) and represent substantial effect sizes given that TVD ranges from 0 to 1. The model as a whole explains approximately 20% of the variance in TVD ($R^2 = 0.202$), which is substantial for a dataset of this size and complexity.

The opposite signs of these coefficients provide strong evidence for our hypothesis that base models and instruction-tuned models respond differently to the challenge of simulating populations with diverse opinions. This pattern holds even after controlling for the specific

datasets and models involved, suggesting it represents a general property of the two model classes rather than an artifact of particular model or evaluation datasets.

Table 5: TVD for each model in SimBenchPop and SimBenchGrouped. Lower values indicate better performance. PT and IT refer to pretrained and instruction-tuned versions, respectively.

| Model | SimBenchPop | SimBenchGrouped | Average |
|---|---|---|---|
| *Baselines* | | | |
| Random baseline | 0.390 | 0.415 | 0.402 |
| Uniform baseline | 0.335 | 0.362 | 0.348 |
| *Commercial Models* | | | |
| Claude-3.7-Sonnet | 0.197 | 0.184 | 0.191 |
| Claude-3.7-Sonnet-4000 | 0.201 | 0.188 | 0.195 |
| GPT-4.1 | 0.212 | 0.205 | 0.209 |
| o4-mini-high | 0.235 | 0.214 | 0.225 |
| o4-mini-low | 0.234 | 0.216 | 0.230 |
| *Open Models* | | | |
| DeepSeek-V3-0324 | 0.215 | 0.218 | 0.216 |
| DeepSeek-R1 | 0.211 | 0.212 | 0.211 |
| Llama-3.1-8B-Instruct | 0.321 | 0.318 | 0.320 |
| Llama-3.1-70B-Instruct | 0.277 | 0.247 | 0.263 |
| Llama-3.1-405B-Instruct | 0.237 | 0.225 | 0.231 |
| Qwen2.5-0.5B | 0.337 | 0.364 | 0.349 |
| Qwen2.5-1.5B | 0.321 | 0.324 | 0.322 |
| Qwen2.5-3B | 0.300 | 0.327 | 0.313 |
| Qwen2.5-7B | 0.290 | 0.326 | 0.307 |
| Qwen2.5-14B | 0.285 | 0.314 | 0.298 |
| Qwen2.5-32B | 0.273 | 0.308 | 0.290 |
| Qwen2.5-72B | 0.269 | 0.300 | 0.283 |
| Gemma-3-1B-PT | 0.382 | 0.413 | 0.396 |
| Gemma-3-4B-PT | 0.334 | 0.342 | 0.338 |
| Gemma-3-12B-PT | 0.310 | 0.317 | 0.314 |
| Gemma-3-27B-PT | 0.309 | 0.325 | 0.317 |
| Gemma-3-4B-IT | 0.337 | 0.341 | 0.339 |
| Gemma-3-12B-IT | 0.262 | 0.274 | 0.267 |
| Gemma-3-27B-IT | 0.270 | 0.273 | 0.272 |

# G   Dataset Details

We provide details on each of the 20 datasets in SimBench. Note that for many datasets we use only a subset of questions and participants for SimBench, as a result of our preprocessing steps (§2.2).

## G.1   WisdomOfCrowds

**Description**: This dataset contains **factual questions** that were administered to a large number of US-based Amazon Mechanical Turk workers. The data was originally collected to study wisdom of the crowd effects.

**Questions**: 113, with an average of 518 responses per question.

**Example question**:

Table 6: Results: Ordinary least squares

| Model: | OLS | Adj. R-squared: | 0.201 |
|---|---|---|---|
| Dependent Variable: | Total_Variation | AIC: | -134342.8438 |
| Date: | 2025-05-15 20:27 | BIC: | -133890.3555 |
| No. Observations: | 172008 | Log-Likelihood: | 67216. |
| Df Model: | 44 | F-statistic: | 983.5 |
| Df Residuals: | 171963 | Prob (F-statistic): | 0.00 |
| R-squared: | 0.201 | Scale: | 0.026805 |

| | Coef. | Std.Err. | t | P> |t| | [0.025 | 0.975] |
|---|---|---|---|---|---|---|
| Intercept | 0.1824 | 0.0029 | 62.1882 | 0.0000 | 0.1766 | 0.1881 |
| C(dataset_name)[T.ChaosNLI] | -0.0442 | 0.0021 | -20.7195 | 0.0000 | -0.0483 | -0.0400 |
| C(dataset_name)[T.Choices13k] | -0.1016 | 0.0021 | -47.3233 | 0.0000 | -0.1058 | -0.0974 |
| C(dataset_name)[T.ConspiracyCorr] | -0.0452 | 0.0052 | -8.6565 | 0.0000 | -0.0554 | -0.0349 |
| C(dataset_name)[T.DICES] | -0.0254 | 0.0023 | -11.0298 | 0.0000 | -0.0300 | -0.0209 |
| C(dataset_name)[T.ESS] | -0.0202 | 0.0021 | -9.4882 | 0.0000 | -0.0244 | -0.0160 |
| C(dataset_name)[T.GlobalOpinionQA] | -0.0428 | 0.0021 | -20.2041 | 0.0000 | -0.0469 | -0.0386 |
| C(dataset_name)[T.ISSP] | -0.0279 | 0.0021 | -13.1516 | 0.0000 | -0.0321 | -0.0238 |
| C(dataset_name)[T.Jester] | 0.1168 | 0.0033 | 35.9190 | 0.0000 | 0.1104 | 0.1232 |
| C(dataset_name)[T.LatinoBarometro] | -0.0325 | 0.0021 | -15.1931 | 0.0000 | -0.0367 | -0.0283 |
| C(dataset_name)[T.MoralMachine] | -0.0380 | 0.0021 | -17.8607 | 0.0000 | -0.0422 | -0.0339 |
| C(dataset_name)[T.MoralMachineClassic] | -0.1594 | 0.0088 | -18.1961 | 0.0000 | -0.1766 | -0.1422 |
| C(dataset_name)[T.NumberGame] | -0.0821 | 0.0021 | -38.8471 | 0.0000 | -0.0863 | -0.0780 |
| C(dataset_name)[T.OSPsychBig5] | -0.1186 | 0.0026 | -45.0783 | 0.0000 | -0.1238 | -0.1134 |
| C(dataset_name)[T.OSPsychMACH] | -0.0227 | 0.0037 | -6.1522 | 0.0000 | -0.0299 | -0.0155 |
| C(dataset_name)[T.OSPsychMGKT] | -0.1121 | 0.0021 | -52.6066 | 0.0000 | -0.1163 | -0.1080 |
| C(dataset_name)[T.OSPsychRWAS] | 0.0168 | 0.0073 | 2.3068 | 0.0211 | 0.0025 | 0.0311 |
| C(dataset_name)[T.OpinionQA] | -0.1013 | 0.0021 | -47.9196 | 0.0000 | -0.1054 | -0.0972 |
| C(dataset_name)[T.TISP] | -0.0441 | 0.0022 | -20.5072 | 0.0000 | -0.0483 | -0.0399 |
| C(dataset_name)[T.WisdomOfCrowds] | -0.0200 | 0.0035 | -5.7228 | 0.0000 | -0.0268 | -0.0131 |
| C(Model)[T.Claude-3.7-Sonnet-4000] | 0.0038 | 0.0027 | 1.3978 | 0.1622 | -0.0015 | 0.0092 |
| C(Model)[T.DeepSeek-R1] | 0.0133 | 0.0027 | 4.8513 | 0.0000 | 0.0079 | 0.0186 |
| C(Model)[T.DeepSeek-V3-0324] | 0.0177 | 0.0027 | 6.4740 | 0.0000 | 0.0123 | 0.0231 |
| C(Model)[T.GPT-4.1] | 0.0141 | 0.0027 | 5.1557 | 0.0000 | 0.0087 | 0.0195 |
| C(Model)[T.Gemma-3-12B-IT] | 0.0641 | 0.0027 | 23.4327 | 0.0000 | 0.0587 | 0.0694 |
| C(Model)[T.Gemma-3-12B-PT] | 0.3616 | 0.0035 | 104.5204 | 0.0000 | 0.3549 | 0.3684 |
| C(Model)[T.Gemma-3-1B-PT] | 0.4330 | 0.0035 | 125.1390 | 0.0000 | 0.4262 | 0.4398 |
| C(Model)[T.Gemma-3-27B-IT] | 0.0730 | 0.0027 | 26.6890 | 0.0000 | 0.0676 | 0.0784 |
| C(Model)[T.Gemma-3-27B-PT] | 0.3604 | 0.0035 | 104.1666 | 0.0000 | 0.3536 | 0.3672 |
| C(Model)[T.Gemma-3-4B-IT] | 0.1398 | 0.0027 | 51.1034 | 0.0000 | 0.1344 | 0.1451 |
| C(Model)[T.Gemma-3-4B-PT] | 0.3857 | 0.0035 | 111.4826 | 0.0000 | 0.3790 | 0.3925 |
| C(Model)[T.Llama-3.1-405B-Instruct] | 0.0392 | 0.0027 | 14.3206 | 0.0000 | 0.0338 | 0.0445 |
| C(Model)[T.Llama-3.1-70B-Instruct] | 0.0792 | 0.0027 | 28.9426 | 0.0000 | 0.0738 | 0.0845 |
| C(Model)[T.Llama-3.1-8B-Instruct] | 0.1231 | 0.0027 | 45.0170 | 0.0000 | 0.1178 | 0.1285 |
| C(Model)[T.Qwen2.5-0.5B] | 0.3880 | 0.0035 | 112.1256 | 0.0000 | 0.3812 | 0.3947 |
| C(Model)[T.Qwen2.5-1.5B] | 0.3719 | 0.0035 | 107.4976 | 0.0000 | 0.3652 | 0.3787 |
| C(Model)[T.Qwen2.5-14B] | 0.3359 | 0.0035 | 97.0893 | 0.0000 | 0.3292 | 0.3427 |
| C(Model)[T.Qwen2.5-32B] | 0.3248 | 0.0035 | 93.8707 | 0.0000 | 0.3180 | 0.3316 |
| C(Model)[T.Qwen2.5-3B] | 0.3517 | 0.0035 | 101.6583 | 0.0000 | 0.3450 | 0.3585 |
| C(Model)[T.Qwen2.5-72B] | 0.3198 | 0.0035 | 92.4342 | 0.0000 | 0.3130 | 0.3266 |
| C(Model)[T.Qwen2.5-7B] | 0.3409 | 0.0035 | 98.5348 | 0.0000 | 0.3342 | 0.3477 |
| C(Model)[T.o4-mini-high] | 0.0374 | 0.0027 | 13.6575 | 0.0000 | 0.0320 | 0.0427 |
| C(Model)[T.o4-mini-low] | 0.0363 | 0.0027 | 13.2773 | 0.0000 | 0.0310 | 0.0417 |
| C(instruct_flag)[base]:Human_Normalized_Entropy | -0.2628 | 0.0026 | -101.0841 | 0.0000 | -0.2679 | -0.2577 |
| C(instruct_flag)[instruct]:Human_Normalized_Entropy | 0.0929 | 0.0024 | 37.9507 | 0.0000 | 0.0881 | 0.0977 |

| Omnibus: | 21133.651 | Durbin-Watson: | 1.711 |
|---|---|---|---|
| Prob(Omnibus): | 0.000 | Jarque-Bera (JB): | 34296.360 |
| Skew: | 0.862 | Prob(JB): | 0.000 |
| Kurtosis: | 4.346 | Condition No.: | 33 |

> An analogy compares the relationship between two things or ideas to highlight some point of similarity. You will be given pairs of words bearing a relationship, and asked to select another pair of words that illustrate a similar relationship.
>
> Which pair of words has the same relationship as 'Letter : Word'?
>
> (A): Page : Book
> (B): Product : Factory
> (C): Club : People
> (D): Home work : School

**Participants**: 722 US-based Amazon Mechanical Turk workers.

**Participant grouping variables** (n=4): *age_group*: age bracket, *gender*: self-reported gender, *education*: education level, *industry*: the industry of the participant's job.

**Default System Prompt**:

> You are an Amazon Mechanical Turk worker from the United States.

**License**: MIT

**Publication**: Simoiu et al. (2019)

### G.2 Jester

**Description**: This dataset contains **jokes** for which participants provided **subjective judgments** of how funny they found them. The data was originally collected to enable recommender systems and collaborative filtering research.

**Questions**: 136, with an average of 779 responses per question.

**Example question**:

> How funny is the following joke, on a scale of -10 to 10? (-10: not funny, 10: very funny)
>
> How many feminists does it take to screw in a light bulb? That's not funny.
>
> Options:
> (A): 7 to 10
> (B): 3 to 6
> (C): -2 to 2
> (D): -5 to -3
> (E): -10 to -6

**Participants**: 7,669 volunteer participants (sociodemographics unknown) who chose to use the Jester joke recommender website.

**Participant grouping variables**: None. **Default System Prompt**:

> Jester is a joke recommender system developed at UC Berkeley to study social information filtering. You are a user of Jester.

**License**: "Freely available for research use when cited appropriately."

**Publication**: Goldberg et al. (2001)

### G.3 Choices13k

**Description**: This dataset contains a large number of automatically generated **decision-making scenarios** that present participants with two lotteries to choose from. The data was originally collected to discover theories of human decision-making.

**Questions**: 14,568, with an average of 17 responses per question.

**Example question**:

> There are two gambling machines, A and B. You need to make a choice between the machines with the goal of maximizing the amount of dollars received. You will get one reward from the machine that you choose. A fixed proportion of 10% of this value will be paid to you as a performance bonus. If the reward is negative, your bonus is set to $0.
>
> Machine A: $-1.0 with 5.0% chance, $26.0 with 95.0% chance.
> Machine B: $21.0 with 95.0% chance, $23.0 with 5.0% chance.
>
> Which machine do you choose?

**Participants**: 14,711 US-based Amazon Mechanical Turk workers.

**Participant grouping variables**: None.

**Default System Prompt**:

> You are an Amazon Mechanical Turk worker based in the United States.

**License**: "All data are available to the public without registration at github.com/jcpeterson/choices13k".

**Publication**: Peterson et al. (2021)

### G.4 OpinionQA

**Description**:

This dataset contains **survey questions** that ask participants to provide **self-assessments** and **subjective judgments**. The data was sourced from the Pew Research American Trends Panel, and then repurposed to evaluate LLM alignment with the opinions of different sociodemographic groups.

**Questions**: 736, with an average of 5,339 responses per question.

**Example question**:

> How would you describe your household's financial situation?
>
> (A): Live comfortably
> (B): Meet your basic expenses with a little left over for extras
> (C): Just meet your basic expenses
> (D): Don't even have enough to meet basic expenses
> (E): Refused

**Participants**: [roughly 10,000] paid participants from a representative sample of the US populace.

**Participant grouping variables** (n=13): *CREGION*: U.S. region of residence, *AGE*: age bracket of the respondent, *SEX*: male or female, *EDUCATION*: highest level of education completed, *CITIZEN*: the respondent is (not) a citizen of the US, *MARITAL*: current marital status, *RELIG*: religious affiliation, *RELIGATTEND*: frequency of religious service attendance,

*POLPARTY*: political party affiliation, *INCOME*: income bracket, *POLIDEOLOGY*: political ideology (e.g., liberal/conservative), *RACE*: racial identity.

**Default System Prompt**:

> You are from the United States.

**License**: No licensing information provided; Data is freely available without registration at https://worksheets.codalab.org/worksheets/0x6fb693719477478aac73fc07db333f69

**Publication**: Santurkar et al. (2023)

## G.5 MoralMachineClassic

**Description**: This dataset contains three **moral decision-making scenarios**, which a large number of participants were asked to provide **subjective choices** for. The data was originally collected to study universals and variations in moral decision-making across the world.

**Questions**: 3, with an average of 17,720 responses per question.

**Example question**:

> A man in blue is standing by the railroad tracks when he notices an empty boxcar rolling out of control. It is moving so fast that anyone it hits will die. Ahead on the main track are five people. There is one person standing on a side track that doesn't rejoin the main track. If the man in blue does nothing, the boxcar will hit the five people on the main track, but not the one person on the side track. If the man in blue flips a switch next to him, it will divert the boxcar to the side track where it will hit the one person, and not hit the five people on the main track. What should the man in blue do?

**Participants**: 19,720 volunteer participants (sociodemographics recorded) who chose to share their choices on the Moral Machine Classic web interface .

**Participant grouping variables** (n=6): *country*: respondent's country of residence, *gender*: gender of the respondent, *education*: level of education, *age_group*: age bracket, *political_group*: self-identified political orientation, *religious_group*: self-identified religious affiliation.

**Default System Prompt**:

> The Moral Machine website (moralmachine.mit.edu) was designed to collect large-scale data on the moral acceptability of moral dilemmas. You are a user of the Moral Machine website.

**License**: No licensing information provided.

**Publication**: Awad et al. (2020)

## G.6 ChaosNLI

**Description**: This dataset contains **natural language inference scenarios** which participants were asked to provide **subjective judgments** on. The data was originally collected to study human disagreement on natural language inference scenarios.

**Questions**: 4,645, with exactly 100 responses per question.

**Example question**:

> Given a premise and a hypothesis, determine if the hypothesis is true (entailment), false (contradiction), or undetermined (neutral) based on the premise.

> Premise: Two young children in blue jerseys, one with the number 9 and one with the number 2 are standing on wooden steps in a bathroom and washing their hands in a sink.
> Hypothesis: Two kids at a ballgame wash their hands.
>
> Choose the most appropriate relationship between the premise and hypothesis:
> (A): Entailment (the hypothesis must be true if the premise is true)
> (B): Contradiction (the hypothesis cannot be true if the premise is true)
> (C): Neutral (the hypothesis may or may not be true given the premise)

**Participants**: 5,268 Amazon Mechanical Turk workers.

**Participant grouping variables**: None.

**Default System Prompt**:

> You are an Amazon Mechanical Turk worker.

**License**: CC BY-NC 4.0

**Publication**: Nie et al. (2020)

### G.7  European Social Survey (ESS)

**Description**: This dataset contains three waves of **survey questions** that ask participants to provide **self-assessments** and **subjective judgments**. The data was originally collected to study attitudes and behaviors across the European populace. We use ESS wave 8-10.

**Questions**: 237, with an average of 41,540 responses per task.

**Example question**:

> Sometimes the government disagrees with what most people think is best for the country. Which one of the statements on this card describes what you think is best for democracy in general?
>
> Options:
> (A): Government should change its policies
> (B): Government should stick to its policies
> (C): It depends on the circumstances

**Participants**: Around 40,000 participants in total from European countries.

**Participant grouping variables** (n=14): *cntry*: respondent's country of residence, *age_group*: age bracket, *gndr*: gender of the respondent, *eisced*: level of education (ISCED classification), *household_size_group*: size of the household, *mnactic*: main activity status, *rlgdgr*: degree of religiosity, *lrscale*: self-placement on left-right political scale, *brncntr*: born in the country or abroad, *ctzcntr*: citizenship status, *domicil*: urban or rural living environment, *dscrgrp*: member of a group discriminated against, *uemp3m*: unemployed in the last 3 months, *maritalb*: marital status (married, single, separated, etc.)

**Default System Prompt**:

> The year is {survey year}.

**License**: CC BY-NC-SA 4.0

**Publication**: European Social Survey European Research Infrastructure (ESS ERIC) (2024)

### G.8 AfroBarometer

**Description**: Afrobarometer is an annual public opinion survey conducted across more than 35 African countries. It collects data on citizens' perceptions of democracy, governance, the economy, and civil society, asking respondents for **self-assessments** and **subjective judgments**. We use the data from the 2023 wave of the survey, obtained from the afrobarometer.org website. We use Afrobarometer Round 9.

**Questions**: 213, with an average of 52,900 responses per question.

**Example question**:

> Do you think that in five years' time this country will be more democratic than it is now, less democratic, or about the same?
>
> Options:
> (A): Much less democratic
> (B): Somewhat less democratic
> (C): About the same
> (D): Somewhat more democratic
> (E): Much more democratic
> (F): Refused
> (G): Don't know

**Participants**: 1,200-2,400 per country, 39 countries

**Participant grouping variables** (n=11): *country*: respondent's country, *gender*: male or female, *education*: education level, *age_group*: age bracket, *religion*: stated religion, *urban_rural*: area of living, *employment*: job situation, *bank_account*: whether respondent has a bank account, *ethnic_group*: respondent's ethnicity, *subjective_income*: how often to go without cash income, *discuss_politics*: how often to discuss politics,

**Default System Prompt**:

> The year is {survey year}.

**License**: No explicit language forbidding redistribute.

**Publication**: Afrobarometer (2023)

### G.9 OSPsychBig5

**Description**: This dataset contains a collection of anonymized **self-assessments** from the Big Five Personality Test, designed to evaluate individuals across five core personality dimensions.

**Questions**: 50, with an average of 19,632 responses per question.

**Example question**:

> Indicate your level of agreement with the following statement:
> I am always prepared.
>
> Options:
> (A): Disagree
> (B): Slightly Disagree
> (C): Neutral
> (D): Slightly Agree
> (E): Agree

**Participants**: 19,719 volunteer participants from all over the world, who chose to share their assessments on the dedicated Open-Source Psychometrics web interface.

**Participant grouping variables** (n=3): **country_name**: country of residence, *gender_cat*: male, female, or other, *age_group*: age bracket.

**Default System Prompt**:

> openpsychometrics.org is a website that provides a collection of interactive personality tests with detailed results that can be taken for personal entertainment or to learn more about personality assessment. You are a user of openpsychometrics.org.

**License**: Creative Commons.

**Publication**: None.

## G.10   OSPsychMGKT

**Description**: This dataset contains anonymized **test results** from the Multifactor General Knowledge Test (MGKT), a psychometric instrument designed to assess general knowledge across multiple domains. Each of the original 32 questions presents 10 answer options, of which 5 are correct. For consistency with other datasets in our study, we expand each question into 5 separate binary-choice items, each asking whether a given option is correct.

**Questions**: 320, with an average of 18,644 responses per question.

**Example question**:

> Is "Emily Dickinson" an example of famous poets?
> Choose one:
> (A) Yes
> (B) No

**Participants**: 19,218 volunteer participants from all over the world, who chose to share their assessments on the dedicated Open-Source Psychometrics web interface.

**Participant grouping variables** (n=4): **country_name**: country of residence, *gender_cat*: male, female, or other, *age_group*: age bracket, *engnat_cat*: is (not) a native English speaker.

> openpsychometrics.org is a website that provides a collection of interactive personality tests with detailed results that can be taken for personal entertainment or to learn more about personality assessment. You are a user of openpsychometrics.org.

**License**: Creative Commons.

**Publication**: None.

## G.11   OSPsychMACH

**Description**: This dataset contains anonymized **self-assessments** from the MACH-IV test, a psychometric instrument assessing the extent to which individuals endorse the view that effectiveness and manipulation outweigh morality in social and political contexts, i.e., their endorsement of Machiavellianism.

**Questions**: 20, with an average of 54,974 responses per question.

**Example question**:

> Indicate your level of agreement with the following statement:
> Never tell anyone the real reason you did something unless it is useful to do so.

Options:
(A): Disagree
(B): Slightly disagree
(C): Neutral
(D): Slightly agree
(E): Agree

**Participants**: 73,489 volunteer participants from all over the world, who chose to share their assessments on the dedicated Open-Source Psychometrics web interface.

**Participant grouping variables** (n=18): **country_name**: country of residence, *gender_cat*: male, female, or other, *age_group*: age bracket, *race_cat*: respondent's race, *engnat_cat*: is (not) a native English speaker, *hand_cat*: right-, left-, or both-handed, *education_cat*: level of education, *urban_cat*: type of urban area, *religion_cat*: stated religion, *orientation_cat*: sexual orientation, *voted_cat*: did (not) vote at last elections, *married_cat*: never, currently, or previously married, *familysize*: number of people belonging to the family, *TIPI_E_Group*: extraversion level based on TIPI score, *TIPI_A_Group*: agreeableness level based on TIPI score, *TIPI_C_Group*: conscientiousness level based on TIPI score, *TIPI_ES_Group*: emotional stability level based on TIPI score, *TIPI_O_Group*: openness-to-experience level based on TIPI score.

> openpsychometrics.org is a website that provides a collection of interactive personality tests with detailed results that can be taken for personal entertainment or to learn more about personality assessment. You are a user of openpsychometrics.org.

**License**: Creative Commons.

**Publication**: None.

### G.12   OSPsychRWAS

**Description**: This dataset contains anonymized **self-assessments** from the Right-Wing Authoritarianism Scale (RWAS), a psychometric instrument assessing authoritarian tendencies such as submission to authority, aggression toward outgroups, and adherence to conventional norms.

**Questions**: 22, with an average of 6,918 responses per question.

**Example question**:

> Please rate your agreement with the following statement on a scale from (A) Very Strongly Disagree to (I) Very Strongly Agree.
>
> Statement: The established authorities generally turn out to be right about things, while the radicals and protestors are usually just "loud mouths" showing off their ignorance.
>
> Options:
> (A): Very Strongly Disagree
> (B): Strongly Disagree
> (C): Moderately Disagree
> (D): Slightly Disagree
> (E): Neutral
> (F): Slightly Agree
> (G): Moderately Agree
> (H): Strongly Agree
> (I): Very Strongly Agree

**Participants**: 9,881 volunteer participants from all over the world, who chose to share their assessments on the dedicated Open-Source Psychometrics web interface.

**Participant grouping variables** (n=18): *age_group*: age bracket, *gender_cat*: male or female or other, *race_cat*: respondent's race, *engnat_cat*: is (not) English native, *hand_cat*: right/left/both-handed, *education_cat*: level of education, *urban_cat*: type of urban area, *religion_cat*: stated religion, *orientation_cat*: sexual orientation, *voted*: did (not) vote at last elections, *married*: never/currently/previously, *familysize*: number of people belonging to the family, *TIPI_E_Group*: extraversion level based on TIPI score, *TIPI_A_Group*: agreeableness level based on TIPI score, *TIPI_C_Group*: conscientiousness level based on TIPI score, *TIPI_ES_Group*: emotional stability level based on TIPI score, *TIPI_O_Group*: openness-to-experience level based on TIPI score. *household_income*: income sufficiency, *work_status*: job situation, *religion*: stated religion, *nr_of_persons_in_household*: 1-7+, *marital_status* respondent's legal relationship status, *domicil*: type of urban area,

> openpsychometrics.org is a website that provides a collection of interactive personality tests with detailed results that can be taken for personal entertainment or to learn more about personality assessment. You are a user of openpsychometrics.org.

**License**: Creative Commons.

**Publication**: None.

### G.13   International Social Survey Programme (ISSP)

**Description**: The International Social Survey Programme (ISSP) is a **cross-national** collaborative programme conducting **annual surveys** on diverse **topics relevant to social sciences** since 1984. Of all 37 surveys, here we include only the five most recent surveys, which were collected in the years 2017 to 2021.

**Questions**: 1,688, with an average of 7,074 responses per question.

**Participants**: 1,000 - 1,500 per country per wave

**Participant grouping variables** (n=11): *country*: respondent's country, *age*: age bracket, *gender*: male or female, *years_of_education*: 1-10+, *household_income*: income sufficiency, *work_status*: job situation, *religion*: stated religion, *nr_of_persons_in_household*: 1-7+, *marital_status* respondent's legal relationship status, *domicil*: type of urban area, *topbot*: self-asessed social class

**Default System Prompt**:

> The timeframe is {survey timeframe}.

**License**: "Data and documents are released for academic research and teaching."

**Publication**: see wave-specific references below.

#### G.13.1   ISSP 2017 Social Networks and Social Resources

**Example question**:

> This section is about who you would turn to for help in different situations, if you needed it.
>
> For each of the following situations, please tick one box to say who you would turn to first. If there are several people you are equally likely to turn to, please tick the box for the one you feel closest to.
>
> Who would you turn to first to help you around your home if you were sick and had to stay in bed for a few days?
>
> Options:

(A): Close family member
(B): More distant family member
(C): Close friend
(D): Neighbour
(E): Someone I work with
(F): Someone else
(G): No one
(H): Can't choose

**Publication**: ISSP Research Group (2019)

### G.13.2 ISSP 2018 Religion IV

**Example question**:

Please indicate which statement below comes closest to expressing what you believe about God.

Options:
(A): I don't believe in God
(B): Don't know whether there is a God and no way to find out
(C): Don't believe in a personal God, but in a Higher Power
(D): Find myself believing in God sometimes, but not at others
(E): While I have doubts, I feel that I do believe in God
(F): I know God really exists and have no doubts about it
(G): Don't know

**Publication**: ISSP Research Group (2020)

### G.13.3 ISSP 2019 Social Inequality V

**Example question**:

Looking at the list below, who do you think should have the greatest responsibility for reducing differences in income between people with high incomes and people with low incomes?

Options:
(A): Cant choose
(B): Private companies
(C): Government
(D): Trade unions
(E): High-income individuals themselves
(F): Low-income individuals themselves
(G): Income differences do not need to be reduced

**Publication**: ISSP Research Group (2022)

### G.13.4 ISSP 2020 Environment IV

**Example question**:

In the last five years, have you ...

Taken part in a protest or demonstration about an environmental issue?

Options:

> (A): Yes, I have
> (B): No, I have not

**Publication**: ISSP Research Group (2023)

### G.13.5   ISSP 2021 Health and Health Care II

**Example question**:

> During the past 12 months, how often, if at all, have you used the internet to look for information on the following topics?
>
> Information related to anxiety, stress, or similar problems?
>
> Options:
> (A): Can't choose
> (B): Never
> (C): Seldom
> (D): Sometimes
> (E): Often
> (F): Very often

**Publication**: ISSP Research Group (2024)

## G.14   LatinoBarómetro

**Description**:

Latinobarómetro is an annual public opinion survey conducted across 18 Latin American countries. It gathers data on the state of democracies, economies, and societies in the region, asking for **self-assessments** and **subjective judgments**. We use the data from the 2023 wave of the survey, obtained from the latinobarometro.org website.

**Questions**: 155, with an average of 18,083 responses per question.

**Example question**:

> Generally speaking, would you say you are satisfied with your life? Would you say you are...
>
> (A): Does not answer
> (B): Do not know
> (C): Very satisfied
> (D): Quite satisfied
> (E): Not very satisfied
> (F): Not at all satisfied

**Participants**: In total, 19,205 interviews were applied in 17 countries. Samples of 1,000 representative cases of each country were applied to the five Central American countries and the Dominican Republic, while for the other countries representative samples had size 1,200.

**Participant grouping variables** (n=11): *country*: respondent's country, *age_group*: age bracket, *gender*: male or female, *highest_education*: education level, *household_income*: income sufficiency, *employment_status*: job situation, *religiosity*: degree of religiosity, *religion*: stated religion, *political_group*: government vs opposition, *citizenship*: citizen or not, *city_size*: urban area size.

**Default System Prompt**:

> The year is {survey year}.

**License**: No explicit language forbidding redistribute.

**Publication**: Latinobarómetro (2023)

### G.15 GlobalOpinionQA

**Description**: This dataset contains survey questions that ask participants to provide **self-assessments** and **subjective judgments**, covering topics such as democracy, governance, international relations, and social values. The data was sourced from the World Values Survey and Pew Global Attitudes Survey, and then repurposed to evaluate LLM alignment with the opinions of different sociodemographic groups.

**Questions**: 2,556, grouping detail unknown

**Example question**:

> Question: Do you strongly agree, agree, disagree, or strongly disagree with the following statement:
> "On the whole, men make better business executives than women do."
>
> (A) Agree strongly
> (B) Agree
> (C) Disagree
> (D) Strongly disagree
> (E) Don't know

**Participants**: Pew Global Attitudes Survey: around 1,000 per country; World Values Survey: around 1,200 per country

**Participant grouping variables** (n=1): country

**Default System Prompt**: None.

**License**: CC BY-NC-SA 4.0

**Publication**: Durmus et al. (2024)

### G.16 Diversity in Conversational AI Evaluation for Safety (DICES) - 990

**Description**: This dataset comprises 990 multi-turn adversarial conversations between humans and a dialogue model, accompanied by raters' **judgement** on the safety of last model response. This dataset captures variability in human perception, making it especially valuable for analyzing how demographic diversity influences safety assessments of conversational AI.

**Questions**: 27,748 (990 turn annotated in multiple dimensions), with an average of 72 responses per question.

**Example question**:

> Conversation:
> A: :)
> B: Are you going to be okay?
> A: Yes I am fine
> B: Oh, well that's good to hear. Glad it all worked out.
> A: Yes
>
> Bot's response: You don't have to talk to me if you don't want to though.

> Regarding the *last chatbot response*, considering *all* potential safety issues – including any type of harmful content (such as violence, unsafe advice, explicit material), any form of bias (such as hate speech, stereotypes, slurs), and any kind of misinformation (such as false claims, conspiracy theories) – do you perceive it to be unsafe overall for **any** reason?
>
> (A): Yes
> (B): No
> (C): Unsure

**Participants**: 173 raters balanced by gender and country (US, India)

**Participant grouping variables** (n=4): *locale*: respondent's country (U.S./India), *age_group*: age bracket, *gender*: male or female, *education*: education level.

> You are a crowd worker.

**License**: CC BY-NC-SA 4.0

**Publication**: Aroyo et al. (2023)

### G.17 NumberGame

**Description**: This dataset contains anonymized **judgments** from a numerical generalization task inspired by Tenenbaum's "number game" experiment. Responses reflect both rule-based (e.g., "even numbers") and similarity-based (e.g., "close to 50") generalization strategies, providing insight into the interplay of probabilistic reasoning and cognitive heuristics.

**Questions**: 25,499, with an average of 10.15 responses per question.

**Example question**:

> A program produces the following numbers: 63_ 43.
>
> Is it likely that the program generates this number next: 24?
> (A): Yes
> (B): No

**Participants**: 575 participants from the U.S.

**Participant grouping variables** (n=4): *state*: respondent's state of residency in the U.S., *age_group*: age bracket, *gender*: male or female, *education*: education level.

> You are an Amazon Mechanical Turk worker from the United States.

**License**: CC0 1.0.

**Publication**: Bigelow & Piantadosi (2016)

### G.18 ConspiracyCorr

**Description**: This dataset contains **judgments** measuring individual endorsement of 11 widely circulated conspiracy theory beliefs.

**Questions**: 9, with an average of 26,416 responses per question.

**Example question**:

> Would you say the following statement is true or false?
>
> Statement: The US Government knowingly helped to make the 9/11 terrorist attacks happen in America on 11 September, 2001
>
> Options:
> (A): Definitely true
> (B): Probably true
> (C): Probably false
> (D): Definitely false
> (E): Don't know

**Participants**: 26,416 participants from 20 different countries.

**Participant grouping variables** (n=4): *Country*: country of origin, *Age_Group*: age bracket of the respondent, *Gender*: gender of the respondent, *Gender*: highest level of education completed

> The year is {survey year}.

**License**: CC0 1.0 Universal.

**Publication**: Enders et al. (2024)

### G.19 MoralMachine

**Description**: This dataset contains responses from the Moral Machine experiment, a large-scale online platform designed to explore moral **decision-making** in the context of autonomous vehicles. Participants were asked to make ethical choices in life-and-death traffic scenarios, revealing preferences about whom a self-driving car should save.

**Questions**: 2,073, with an average of 4,601 responses per question.

**Example question**:

> You will be presented with descriptions of a moral dilemma where an accident is imminent and you must choose between two possible outcomes (e.g., 'Stay Course' or 'Swerve'). Each outcome will result in different consequences. Which outcome do you choose?
>
> Options:
>
> (A): Stay, outcome: in this case, the self-driving car with sudden brake failure will continue ahead and drive through a pedestrian crossing ahead. This will result in the death of the pedestrians.
> Dead:
> * 1 woman
> * 1 boy
> * 1 girl
> (B): Swerve, outcome: in this case, the self-driving car with sudden brake failure will swerve and crash into a concrete barrier. This will result in the death of the passengers.
> Dead:
> * 1 woman
> * 1 elderly man
> * 1 elderly woman

**Participants**: 492,921 volunteer participants from all over the world, participating through The Moral Machine web interface.

**Participant grouping variables** (n=1): *UserCountry3*: participant country,

The Moral Machine website (moralmachine.mit.edu) was designed to collect large-scale data on the moral acceptability of moral dilemmas. You are a user of the Moral Machine website.

**License**: No formal open license is declared. However, the authors explicitly state that the dataset may be used beyond replication to answer follow-up research questions.

**Publication**: Awad et al. (2018)

### G.20 Trust in Science and Science-Related Populism (TISP)

**Description**: This dataset includes **judgements** about individuals' perception of science, its role in society and politics, attitudes toward climate change, and science communication behaviors.

**Questions**: 97, with an average of 69.234 responses per question.

**Example question**:

How concerned or not concerned are most scientists about people's wellbeing?

Options:
(A): not concerned
(B): somewhat not concerned
(C): neither nor
(D): somewhat concerned
(E): very concerned

**Participants**: 71,922 participants across 68 countries.

**Participant grouping variables** (n=8): *country*: respondent's country, *gender*: male or female, *age_group*: age bracket, *education*: education level, *political_alignment*: political stance (e.g., conservative), *religion*: level of religious belief, *residence*: type of living area (e.g., urban, rural), *income_group*: income bracket.

The year is {survey year}.

**License**: no explicit language forbidding redistribute.

**Publication**: Mede et al. (2025)

## H    Additional Related Work

**Opinion Benchmarks**    Most closely related to SimBench are OpinionQA (Santurkar et al., 2023) and GlobalOpinionQA (Durmus et al., 2024), which evaluate how accurately LLMs represent viewpoints of specific demographic groups. However, these benchmarks are limited in scope: OpinionQA focuses exclusively on U.S. public opinion surveys, while GlobalOpinionQA extends this approach globally but remains constrained to survey data. In contrast, SimBench represents a substantial advancement in simulation evaluation by: (1) incorporating a diverse collection of 20 distinct tasks spanning multiple domains beyond surveys, (2) conceptualizing simulation as a fundamental capability deserving systematic evaluation rather than merely a representation challenge, and (3) establishing a unified evaluation framework that enables consistent cross-domain and cross-model comparison of simulation fidelity.

**Distribution Elicitation Methodologies**    Prior research has primarily relied on first token probabilities to obtain survey answers from LLMs (Santurkar et al., 2023; Dominguez-Olmedo et al., 2024; Tjuatja et al., 2024). Unlike typical language model applications that

focus on the model's most likely completion, group-level LLM simulations aim to obtain normalized probabilities across all answer options. Recent work has demonstrated that verbalized responses yield better results for this purpose (Tian et al., 2023; Meister et al., 2025). Nevertheless, calibration of LLM outputs remains an open challenge; while extensively studied for model answer confidence (Zhao et al., 2021; Jiang et al., 2021; Kapoor et al., 2024; Zhu et al., 2023) and hallucinations (Kalai & Vempala, 2024), these issues also apply to simulating population response distributions. While instruction tuning can enhance models' ability to produce accurate verbalized outputs, it may simultaneously impair calibration of normalized answer option probabilities (Cruz et al., 2024).

**Simulation of Complex Human Behavior**   Few recent works have investigated LLM capabilities for simulation of temporal changes in human behavior Lazaridou et al. (2021). Ahnert et al. (2024) propose temporal adapters for LLMs for longitudinal analysis. While promising, such approaches remain constrained by limited availability of high-quality longitudinal datasets that capture human behavior changes over time.

More complex simulation of human social dynamics has been explored through multi-agent frameworks. Park et al. (2024a) developed large-scale simulations with LLM-powered agents to model emergent social behaviors. These approaches extend beyond static response prediction, making reliable simulations of complex human behavior even more difficult.

# I   Implementation Details

For base models, we use HuggingFace Transformers (Wolf et al., 2020) to run inference on a single NVIDIA RTX A6000 Ada GPU. We structure prompts so that the next token corresponds to the model's answer choices. For models smaller than 70B parameters, we use 8-bit quantization implemented in bitsandbytes (Dettmers et al., 2022), while 70B models use 4-bit quantization.

For instruction-tuned models, we use API calls. OpenAI models are accessed directly through their API, while other models are accessed via OpenRouter. We request verbalized probability outputs in JSON format with temperature initially set to 0. If parsing fails, we increase temperature to 1 and retry up to 5 times. All models successfully produced valid JSON under these conditions. When probability outputs do not sum to 1, we apply normalization.

Our evaluation includes a diverse set of models: Qwen 2.5 (Yang et al., 2024) (0.5B-72B), Gemma 3 (Team et al., 2025) PT and IT (4B-27B), o4-mini (OpenAI, 2025b), Claude 3.7 Sonnet (Anthropic, 2025), DeepSeek R1 (Guo et al., 2025), DeepSeek-V3-0324 DeepSeek-AI (2024), GPT-4.1 OpenAI (2025a), and Llama-3.1-Instruct (8B-405B) (Meta AI, 2024).

To ensure the validity of our results, we perform two checks: 1) We verify that base models assign the vast majority of probability mass to the provided answer options. Even for small models like Qwen2.5-0.5B, the sum of probabilities across answer tokens is as high as 0.98, confirming that models rarely predict tokens outside the designated answer space. 2) We also evaluate the effect of quantization on model performance using a subset of SimBench. As shown in Table 7, performance remains consistent across quantization levels, with minimal variation in total variation scores even for quantization-sensitive models like Llama-3.1.

We detail below the prompts used in our experimental conditions for token probability and verbalized distribution prediction.

The following system prompt was consistent across all experimental conditions:

> You are a group of individuals with these shared characteristics:
> {default system prompt}{grouping system prompt (if any)}

For token probability prediction, we adapted the prompt structure from Nori et al. (2023):

```
**Question**: {question}
Do not provide any explanation, only answer with one of the following options: {answer
↪    options}.
**Answer**: (
```

Prompt for eliciting verbalized probability prediction:

```
**Question**: {question}
Estimate what percentage of your group would choose each option. Follow these rules:
1. Use whole numbers from 0 to 100
2. Ensure the percentages sum to exactly 100
3. Only include the numbers (no % symbols)
4. Use this exact valid JSON format: {answer options} and do NOT include anything else.
5. Only output your final answer and nothing else. No explanations or intermediate steps
↪    are needed.
Replace X with your estimated percentages for each option.
'**Answer**:
```

Table 7: Total Variation for different models at various quantization levels. Lower values indicate better performance.

| Model | 4-bit | 8-bit | 16-bit | 32-bit |
|---|---|---|---|---|
| Llama-3.1-8B-Instruct | 0.272 | 0.266 | 0.262 | 0.262 |
| Qwen2.5-7B | 0.307 | 0.307 | 0.306 | 0.307 |

