# OpenReview forum: "SimBench: Benchmarking the Ability of Large Language Models to Simulate Human Behaviors"
_colmweb.org/COLM/2025/Workshop/Social_Sim — Social Sim'25_

### Official Review · Reviewer_nEuV · 2025-07-15

**Rating:** 7
**Overall Assessment:** 3
**Confidence:** 5

**Review:**

see above

**Comments Suggestions And Typos:**

See above.

**Ethical Concerns:**

No ethical concerns.

**Paper Summary:**

The paper introduces SimBench, a large-scale benchmark designed to evaluate the ability of LLMs to simulate group-level human behaviors across diverse tasks and demographic groups. The authors compile 20 datasets into a unified format. They have several findings revealing the current limitations of different LLMs in simulation and offer an outlook for future work.

**Relevance:**

5

**Summary Of Strengths:**

The benchmark integrates 20 diverse datasets, enabling evaluation across a wide range of human behaviors and demographic contexts.

A newly proposed sim score is a valuable metric for evaluating the distances of LLM responses und real human response distributions.

Sound experimentation.

**Summary Of Weaknesses:**

Inconsistent methods have been used to extract the model outputs. For base models they use token probabilities and for instruction-tuned models, they use verbalized distributions. This is not consistent.

There is no exploration of advanced techniques (e.g., few-shot prompting, cot, ...) that could improve simulation fidelity.

The benchmark focuses exclusively on multiple-choice, single-turn questions, ignoring more complex, dynamic, or interactive human behaviors.

While the results highlight LLMs’ shortcomings, the paper does not investigate why these failures occur. A deeper analysis would have strengthened the work.

---

### Official Review · Reviewer_akpq · 2025-07-18

**Rating:** 9
**Overall Assessment:** 4
**Confidence:** 4

**Review:**

**Quality**: The paper is of high quality. The methodology is sound, and the analysis is thorough.

**Clarity**: The writing is clear and well-organized. The paper is easy to follow.

**Originality**: The work is original. A large-scale, unified benchmark for a specific purpose is a novel and valuable contribution.

**Significance**: As the interest in using LLMs to model human behavior grows, a standardized and rigorous benchmark is crucial.

**Comments Suggestions And Typos:**

* For improved readability, consider introducing the core evaluation metrics and measurements in the introduction. This would give readers a clear framework for interpreting the experimental results as they are presented throughout the paper.
* I suggest increasing the font size in Figure 4.

**Paper Summary:**

The authors introduce SimBench, a comprehensive benchmark designed to evaluate how well Large Language Models (LLMs) can simulate group-level human behaviors. The work compiles 20 diverse datasets into a unified format, capturing a wide range of behaviors from hundreds of thousands of participants globally. The primary goal is to measure the ability of LLMs to simulate distinct types of human behavior and to understand when, how, and why these simulations succeed or fail. Key findings include a clear log-linear scaling relationship between simulation accuracy and model size, as well as a strong correlation between a model's simulation performance and its scientific reasoning capabilities.

**Relevance:**

5

**Summary Of Strengths:**

* The consolidation of 20 different datasets, representing a vast and diverse participant pool, provides a robust basis for evaluating the generalizability of LLM simulators.

* The paper is very well-written and transparent in its methodology and reporting of results.

* The evaluation is comprehensive, testing a wide array of prominent LLMs.

**Summary Of Weaknesses:**

* The benchmark is constrained to a multiple-choice question format. It may not fully capture the complexity and nuance of human behavior, which often involves open-ended responses and actions.
* The use of logits to determine model preferences might introduce model-specific biases.
* The benchmark appears to be primarily in English, with datasets from other languages being translated. This process may introduce cultural or linguistic biases.

---

### Meta-Review · Area_Chair_CPW5 · 2025-07-21

**Recommendation:** Accept

**Metareview:**

--